# Responding to Urinary Tract Infection Symptoms in England’s Community Pharmacies

**DOI:** 10.3390/antibiotics12091383

**Published:** 2023-08-30

**Authors:** Sejal Parekh, Kieran Hand, Lingqian Xu, Victoria Roberts, Fionna Pursey, Diane Ashiru-Oredope, Donna M. Lecky

**Affiliations:** 1Primary, Community and Personalised Care Directorate, NHS England, London SE1 8UG, UK; 2AMR Programme, Medical Directorate, NHS England, Leeds LS2 7UE, UK; 3HCAI, Fungal, AMR, AMU & Sepsis Division, UK Health Security Agency, London SW1P 3HX, UKdiane.ashiru-oredope@ukhsa.gov.uk (D.A.-O.);

**Keywords:** urinary tract infections, community pharmacy, TARGET Toolkit, pharmacy quality scheme, antimicrobial stewardship, medication safety, incentivisation, primary care, antimicrobial resistance

## Abstract

Most urinary tract infections (UTIs) are self-limiting and frequently present in primary care; it is common for patients to seek symptom relief. The TARGET Treating Your Infection (TYI) leaflet was used to respond to UTI symptoms for women under 65 years presenting in community pharmacies. The widespread use of these leaflets was incentivised as part of NHS England’s Pharmacy Quality Scheme (PQS) 2022–23, between October 2022 and March 2023. The TARGET TYI leaflets are aimed to support appropriate antibiotic use and antimicrobial stewardship (AMS) as well as reducing the opportunity for resistance to develop. A total of 8363 community pharmacies completed the AMS criteria within the PQS and collectively submitted data for 104,142 patients presenting with UTI symptoms. The majority, 77% (75,071), of (non-pregnant) women presented with none or only one of the three strongly predictive symptoms of dysuria, new nocturia, cloudy urine, and/or vaginal discharge and, therefore, were less likely to have a UTI, as outlined in the English UTI diagnostic guidance. Conversely, 23% (22,381) of women presented with two or more symptoms of dysuria, new nocturia, cloudy urine, and with no vaginal discharge and, therefore, they were more likely to have a UTI. The TARGET TYI UTI leaflets support community pharmacy teams to differentiate between symptoms more likely to be associated with UTIs and those that could be managed with self-care. The findings suggest that most women presenting to community pharmacies with urinary symptoms were likely to have self-limiting symptoms, and could be suitably managed with self-care, pain relief, and appropriate safety netting. Approximately one-third of patients were managed by community pharmacy team members without the need for referral to a pharmacist and one in five patients presented with escalation symptoms and were signposted to other healthcare settings. A total of 94% (97,452) of women received self-care advice of which 36% (37,565) were also provided with additional patient information leaflets.

## 1. Introduction

An uncomplicated urinary tract infection (UTI) (often known as cystitis or a lower urinary tract infection) is a bacterial infection of the bladder and associated structures [1]. Typical symptoms include painful voiding (dysuria), increased urgency, frequent urination, suprapubic pain and haematuria [2,3]. They are also one of the most common conditions presenting in primary healthcare with an acute UTI occurring in up to 50% of all women in their lifetime, with further estimates suggesting that by the age of 24 years nearly one third of females will have had at least one episode of cystitis [2,4,5,6]. The higher prevalence in women is thought to be due to anatomical differences including a shorter urethra and a lack of prostatic secretions [7]. *Escherichia coli* strains are typically the cause of infection, originating from faecal flora, colonising the vagina and urethra before infecting the bladder [7].

Most cases of uncomplicated UTIs will resolve without requiring treatment, but many patients seek therapy for symptom relief [1]. This is an opportunity for healthcare professionals to offer safety netting (which is the information given to a patient/carer during a primary care consultation about actions to take if their condition fails to improve, changes or if they have further concerns about their health). When UTIs do not resolve, antibiotic stewardship strategies advocate using narrow spectrum antibiotics for the shortest duration required for clinical recovery. UK clinical guidelines recommend a 3 day course of either nitrofurantoin or trimethoprim (if low risk of resistance) as the first-line antibiotic therapy for lower UTIs in non-pregnant adult women [8]. 

Pharmacists are the third largest regulated professional group in the healthcare workforce, of which community pharmacies are the most represented [9]. In addition, they are the most easily accessible health care profession to the public with approximately 11,000 community pharmacies in England; approximately 89% of the population can access a community pharmacy within a 20 min walk [10]. Pharmacy professionals (pharmacists and technicians) play a crucial role in antimicrobial stewardship (AMS), promoting the responsible and effective use of antimicrobial agents to combat antimicrobial resistance (AMR) and improve patient outcomes. 

The Pharmacy Quality Scheme (PQS) forms part of the community pharmacy contractual framework in England [11,12,13]. It is the first quality scheme for England’s community pharmacies and was introduced in 2016, incentivising community pharmacies to deliver quality criteria in a number of quality dimensions, specifically patient safety/clinical effectiveness, patient experience, healthy living/prevention, and digital enablers [13]. Despite being voluntary, the scheme has had consistently high levels of engagement from community pharmacies from its inception with the majority of community pharmacy contractors participating in all domains [14]. This has allowed clinical progress and improved patient safety to be achieved at pace [15].

Since 2020, the PQS has incentivised the increased role of community pharmacy teams in AMS, adding and renewing initiatives for community pharmacies to help embed them into day-to-day practice [16,17]. In 2022–23, the PQS incentivised the use of the TARGET Treating Your Infection (TYI) leaflets for UTIs and Respiratory Tract Infections (RTIs) for pharmacy staff responding to urinary and respiratory symptoms for walk-in patients [18]. The TARGET TYI leaflets are intended to be used in consultation with patients and the UTI leaflet follows the English national UTI diagnostic algorithm [19,20]. The TARGET TYI UTI leaflet has been previously trialled in GP settings and has been endorsed by the National Institute for Clinical Excellence (NICE) [20,21]. 

These leaflets have been adapted to support the role of the pharmacy team in AMS, to ensure patients receive consistent key messaging across the patient pathway [20,22]. The overarching aim of these initiatives is to promote AMS and decrease antimicrobial resistance. This study focuses on the findings of the implementation of the TARGET TYI UTI patient information leaflet. Findings from the deployment of the RTI leaflet within the PQS will be reported separately.

## 2. Materials and Methods

### 2.1. Study Design

This was a service evaluation of the scaling up of the implementation of the TARGET TYI UTI leaflet for the management of UTIs (Figure 1), included in PQS 2022–23. In this structured observational study, researchers assessed the outcomes of how community pharmacists implemented the use of the TARGET TYI UTI leaflets within the prevention domain of the PQS, using data collected in PQS 2022–23 [14]. The PQS criteria were developed by NHS England in collaboration with the United Kingdom Health Security Agency (UKHSA) and the Pharmaceutical Services Negotiating Committee (PSNC) (now known as Community Pharmacy England). The specific requirement to implement the use of the TARGET TYI leaflets was built on previous AMS activities included in PQS Year 2020–21 and 2021–22 [16,17].

### 2.2. Setting and Participants

The PQS is a voluntary scheme in which all community pharmacies in England providing NHS services are eligible to participate [18]. In March 2023, there were 11,051 registered community pharmacies in England [19]. Participating community pharmacies recorded data from walk-in patients requesting advice for the management of urinary symptoms. Any female patient under 65 years presenting with urinary symptoms was eligible for inclusion, patients not meeting this inclusion criterion were excluded. The follow-up of the outcome of patients signposted to other healthcare settings was outside the scope of this study.

The leaflets have been designed in collaboration with professional medical bodies including the Royal College of General Practitioners (RCGP) for women under 65 years only. (A separate set of resources are available for women over 65 years [23]. Protocols for managing women with UTIs are sub-divided by age as this is characterised as a complicating factor, specifically for women over 65 years of age.) [24,25].

The information requirements and guidance about the criteria within the scheme were communicated by the Department of Health and Social Care (DHSC) via the Drug Tariff and NHS England’s PQS Guidance [13,20]. Additional supporting information for contractors was provided by the PSNC [18]. In October 2022, UKHSA and NHS England distributed 2 × A4 laminated TARGET TYI leaflets (for both UTIs and RTIs) and an orientation flowchart to every community pharmacy on the NHS England pharmaceutical list to support the implementation of the TARGET resources in community pharmacies [26]. The flow chart detailed how and when to use each TARGET resource to facilitate the shared decision-making process. The TARGET TYI UTI leaflets include a section to differentiate between women who are more likely to have UTIs and those who are not, i.e., if they had 2 or more of the following symptoms dysuria, new nocturia, cloudy urine, and no vaginal discharge. (Women were less likely to have a UTI if they had none or only one of the symptoms of dysuria, new nocturia, cloudy urine, and/or vaginal discharge.) The leaflets also emphasise important self-care and safety-netting advice.

### 2.3. Data Collection

Pharmacy teams were asked to use the TARGET TYI UTI leaflet for women under 65 years when responding to symptoms of UTIs for four weeks for 15 patients or up to eight weeks if the number of patients was not achieved within four weeks [8]. Community pharmacy teams could submit their data at eight weeks if they had not reached the sample size. (Pharmacy teams could be comprised of pharmacists, pharmacy technicians, trainee pharmacists, trainee pharmacy technicians, dispensary staff, and medicines counter assistants.) Pharmacy teams were required to submit the data collected via the Manage Your Service (MYS) software portal by either the date of their PQS declaration or no later than 31 March 2023. They could complete the data collection from the launch of the scheme on 10 October 2022 to its closure on 31 March 2023. The NHS Business Services Authority (NHSBSA) produced a digital version of the questions from the TARGET TYI leaflets. These were made available to all pharmacy contractors via the MYS portal for data submission. A single form submission was required per patient. 

### 2.4. Data Analysis

Descriptive analysis was conducted by using R statistical software version 4.2.1 (R Foundation for Statistical Computing) and the results were reported as frequencies (and percentages).

### 2.5. Ethics

As this study is a service evaluation, NHS ethical approval was not required [22]. Further institutional ethical approval was not required as confirmed through the UKHSA Research Ethics and Governance Group and the NHS Health Research Authority Decision tool [22]. These data sets were collected by the NHS Business Services Authority (BSA) and used to undertake the analysis. They were also used as part of provider assurance activity to confirm each community pharmacy had completed the requirement for the PQS as declared. No patient identifiable data were recorded within the data collection. All files were handled in accordance with the Data Protection Act 2018 and the United Kingdom General Data Protection Regulation (GDPR).

## 3. Results

### 3.1. PQS Uptake

Data were collected from 14 October 2022 until the 31 March 2023 with 8363 community pharmacies submitting data for 104,142 TARGET Treating Your UTI leaflets. Of these, 73% (6101) submitted data for the required 15 patients, with the remainder submitting data for fewer than 15 patients. The majority of women presenting in community pharmacies self-reported they were not pregnant, i.e., 93.5% (97,452) of patients, whilst 4.5% (4728) self-reported they were pregnant and 2% (1962) were uncertain/not sure. 

### 3.2. Staff Members and Patient Leaflets

Staff members using the leaflets with patients were: pharmacists undertaking 53,479 (51%) of consultations; pharmacy technicians/dispensers (including trainees), undertaking 25% (25,492) of consultations; counter staff, undertaking 20% (21,214) of consultations; and trainee pharmacists, undertaking 4% (3957) consultations. 

### 3.3. Patient Symptoms

The frequencies of the presenting urinary symptoms are described in Table 1. The most common symptoms patients presented with were dysuria (53%, 55,665), increased frequency of urination (53,609, 51%), and increased urgency to urinate (47%, 49,397). The least common symptoms included haematuria (6%, 285), abnormal vaginal discharge (6%, 5935), and other (2%, 2048), which comprised mainly of back pain (473), vaginal itching (385), and smelly urine (211) (Table 1).

The TARGET TYI UTI leaflets, in line with English national diagnostic algorithms, indicate that two or more out of cloudy urine, nocturia, and dysuria are strongly predictive UTI symptoms in non-pregnant women with no vaginal discharge; 77% (75,071) of women did not meet these criteria. In contrast, 23% (22,381) women met these criteria and potentially required escalation to a GP. In addition, 86% (89,233) of women presented with 1–3 symptoms, with very few women presented with five or more symptoms (1%, 1487) (Table 1).

### 3.4. Patient Signposting

The TARGET TYI UTI leaflets highlight that in certain instances it is necessary for a pharmacist to escalate by signposting the patient to another health care setting such as their general practitioner (GP), an out-of-hours service, or an accident and emergency department for urgent assessment. This may occur when a patient presents two or more symptoms, indicating that they are more likely to have a UTI, or when presenting with additional escalation symptoms as outlined in the TARGET TYI UTI leaflet (under the leaflet heading ‘when should I get help?’) (Table 2).

The main reasons for signposting (Table 2) included symptom deterioration, 29% (6788); blood in urine, 20% (4707); and/or kidney pain in back just below the ribs, 17% (3897). 

Of the 23% (22,381) of patients whose symptoms were strongly predictive of a UTI (two or more of cloudy urine, nocturia, dysuria), 31% (6883) of these were escalated to another health care setting. In contrast, all 22% of the total patients (23,351) who presented with symptoms outlined in the ‘when should I get help?’ symptom section of the leaflet was escalated (Table 3). Additionally, 15% of total patients (15,846) required signposting to a GP (76%, 12,022), an out-of-hours service (15%, 2366), or an accident and emergency department for urgent assessment (2%, 324). Some patients (7%, 1134) were signposted to regional/local services where patients could be assessed and supplied a prescription-only medication (POM) using a Patient Group Direction (PGD). (A PGD provides a legal framework that allows some registered health professionals to supply and/or administer specified medicines to a pre-defined group of patients, without them having to see a prescriber, e.g., doctor.) Women who presented with strongly predictive symptoms, as outlined in the TARGET TYI UTI leaflet, presented with more escalation symptoms—22% (4928) in comparison to all patients (Table 3).

### 3.5. Over the Counter Treatment

Over the counter (OTC) (also known as non-prescription) treatments were recommended to 68% (70,466) and supplied to 60% (62,592) of patients; 8% (7874) of patients declined OTC remedies. Among the 22,381 non-pregnant women with symptoms strongly predictive of a UTI, 38% (8387) were not offered an OTC treatment, while 62% (13,994) purchased an OTC treatment (Table 4) and a further (8%) 1838 women were offered but declined recommended products. 

The most common OTC products recommended were cystitis relief sachets, 76% (53,578), and pain relief, 39% (27,533), whilst cranberry products, 14% (10,034), and D-mannose, (1%) 788, typically used for the prevention or prophylaxis of recurrent UTIs, were less frequently recommended (Table 4). Other medicines supplied included treatment for thrush such as clotrimazole products (2%, 1420) and antibiotics (such as nitrofurantoin via a PGD) (1%,748) (Table 4).

For women with urgent escalation symptoms, 74% (11,785/15846) were not supplied any OTC medicines, 20% (3223/15846) were supplied OTC treatment, and 5% (838/15846) were offered but declined an OTC remedy. In addition, 72% (2926/4061) of women were recommended pain relief and 46% (1867/4061) were recommended and offered cystitis relief sachets (Table 4). 

Many women were managed either by the pharmacist, 39% (40,097)**,** or referred to the pharmacist, 30% (32,008), with approximately one third being managed by wider members of the community pharmacy team, 31% (32,037). Moreover, 94% (97,452) of women received self-care advice, of which 36% (37,565) were also provided with additional patient information leaflets.

## 4. Discussion

A majority of 76% (8363) of community pharmacies in England submitted data for 104,142 patient consultations using TARGET TYI UTI leaflets, of which 73% (6101) submitted data for the required 15 patients, as specified in the PQS criterion. In this study, 77% (75,071) of non-pregnant women presented with either vaginal discharge or none or only one of the symptoms of dysuria, new nocturia, and cloudy urine, and therefore were less likely to have a UTI. In contrast, 23% (22,381) of women presented with two or more symptoms of dysuria, new nocturia, cloudy urine, and with no vaginal discharge and, therefore, they were more likely to have a UTI. Approximately one third of patients were managed without the need for referral to a pharmacist and one in five patients presented with escalation symptoms and were signposted to other healthcare settings such as their GP, an out-of-hours service, local service provision, or an accident and emergency department. In addition, 94% (97,452) of women received self-care advice, with 36% (37,565) being provided with patient information leaflets.

The findings suggest that most women presenting themselves to community pharmacies with urinary symptoms were likely to have self-limiting symptoms, with self-care, pain relief, and appropriate safety netting being a suitable way to manage them. The TARGET TYI UTI leaflets are designed to assist community pharmacy teams to differentiate between symptoms more likely to be associated with UTIs and those that could be managed with self-care and symptom relief. 

### 4.1. Management and Escalation of Patient Presenting with Symptoms That Require Escalation

Community pharmacy teams reported that they escalated 100% of the 15% of women identified in these consultations to have escalation symptoms to various healthcare settings. Some integrated care systems in England, have already begun to commission community pharmacists to assess women and to treat uncomplicated UTIs with antibiotics under the legal authority of patient group directions (PGDs). These services can potentially improve access to treatment for women experiencing these symptoms of UTIs. In addition, community pharmacy teams have undertaken national training and participated in PQS criteria for risk management, sepsis recognition, identifying and assessing the risk of missing red flag symptoms during over the counter (OTC) consultations, AMS and infection control training, and pledging to become antibiotic guardians [16]. This requires members of the community pharmacy team to make a pledge stating how they will make better use of antibiotics and help preserve the effectiveness of these vital medicines (the introduction of the Antibiotic Guardian pledge as part of the PQS in 2020/21 led to a vast increase in UK pledges from community pharmacy teams in 2020, which has been sustained) [16]. From these criteria and the use of the TARGET resources, PQS has supported the education and training of community pharmacy teams in managing women with urinary symptoms by reinforcing messages of self-care where appropriate or referring onwards to the most appropriate NHS service depending on their symptoms and clinical assessment [15,26]. Consultation time pressures combined with late symptom presentation are a challenge for even the most experienced of GPs; therefore, having the TARGET resources available in community pharmacies where patients can get an immediate consultation can help to improve early access to assessment and treatment. [20]. Approximately 25% of women with symptoms that required escalation were recommended OTC remedies. This has the risk of potentially masking symptoms or delaying patients seeking the care they require. Pain relief would be appropriate with counselling on the importance of the patients seeking further advice from a GP or an OOH service.

### 4.2. Symptoms and Management of Uncomplicated UTIs

Many women were offered urine alkalisation products, 55% (53,578), followed by pain relief, 28% (27,433), cranberry products, 10% (10,007), and D-Mannose, 1% (786). However, the use of these OTC medicines is not always consistent with national guidance for the treatment and relief of UTI symptoms [25].

Urine alkalisation products such as potassium and sodium citrate are designed to reduce the acidity of urine and are commonly purchased for the relief of urinary symptoms [27]. They work by increasing urine’s pH with the aim of relieving common symptoms of dysuria and increased urinary frequency [27]. However, efficacy data are lacking [27]. These remedies do encourage fluid intake and can provide validation of illness as well as reassurance to some patients until self-limiting symptoms resolve [20]. There is evidence to suggest many women are open to, and some even prefer, an alternative to antibiotic therapy [20,21].

With regard to pain relief, it is unclear which products were recommended by community pharmacy teams participating in the AMS criteria in PQS. Paracetamol for pain management has a well-established efficacy and safety profile and is therefore more likely to be suitable for most women. It can also be used by pregnant people [25]. Ibuprofen can be used if not contraindicated [25]. 

Cranberry products are often recommended for UTI prophylaxis by reducing the risk of symptomatic, culture-verified UTIs in women with recurrent UTIs, but the current evidence suggests the benefits are likely to be small. [28]. In contrast, D-mannose products appear to show efficacy in reducing not only the incidence of recurrent UTIs and their symptoms, but also prolonging UTI-free periods. D-mannose is a monosaccharide which works by inhibiting bacterial adhesion to the urothelium [29]. Furthermore, evidence suggests that D-mannose is also suitable in managing acute symptoms [30]. D-mannose has a high sugar content and should be used with caution in people where this might be an issue.

In summary, pain relief and D-mannose have proven to be effective in managing patients’ acute symptoms. Cranberry products can help to prevent UTIs in women who suffer from recurrent infections. Other useful self-care advice addresses dehydration as a possible cause of UTIs, where patients should be advised about drinking enough fluids to avoid dehydration [25]. 

### 4.3. Uptake of the Initiative from Community Pharmacies

The majority of community pharmacy contractors in England have not only participated in this PQS, but also in historic schemes [14]. In 2021–22, the AMS criterion included the use of the TARGET Antibiotic Checklist for any patient presenting with an antibiotic prescription to educate them about antibiotics and promote AMS [31]. As a result of these consecutive AMS initiatives, community pharmacy teams can effectively counsel and advise patients presenting with antibiotic prescriptions for dispensing as well as those attending the pharmacy for advice and responding to symptoms in the management of common conditions such as a UTI [17].

For the 2023–24 PQS, all elements of the AMS criteria have been renewed as well as including a new element of an “antibiotic amnesty” where community pharmacy teams are required to educate the public on returning leftover/expired antibiotics [12]. The level of the participation in the initiatives has allowed community pharmacies to spearhead AMS activities as well as promote self-care for patients in primary care since 2020. In addition, some local commissioners have risk assessed, developed, and implemented commissioned antibiotic PGDs for patients with common infections such as UTIs. 

In England, there is a plan to improve access to primary care by launching a new pharmacy service, the Common Conditions Service, (by the end of 2023) to enable patients, when appropriate, to access treatment including prescription-only medicines (POMs) using PGDs for seven common infections, specifically for sinusitis, sore throat, earache, infected insect bite(s), impetigo, shingles, and uncomplicated UTIs (in women). This service, together with the existing NHS oral contraception and blood pressure checks services, is estimated to save 10 million appointments in general practice a year once scaled [32]. 

The PQS has raised awareness of AMS and embedded the use of TARGET toolkit resources such as the TARGET Antibiotic Checklist and TARGET TYI leaflets into everyday practice. It has highlighted that there is a role that community pharmacies can play in AMS, supporting the general public and safely reducing the burden on other primary care services.

### 4.4. Strengths and Limitations 

Much like related PQS studies, the data have a high level of ecological validity, where the focus was to promote AMS and the data collected and analysed were from everyday practice over the duration of the scheme [17,33]. Over 100,000 data entries were collected as part of this PQS from across England, making it one of the largest studies conducted in community pharmacies to date. 

TARGET AMS resources are already embedded in English general practice. By embedding the use of these resources via the PQS, nearly all of community pharmacy teams in England are now also familiar with TARGET Toolkit resources. Patients are, therefore, provided with consistent evidence-based AMS messaging across the primary care pathway. The initiatives have been structured to create a solid foundation of AMS via training and education, the awareness of local formularies and pledging to be antibiotic guardians, followed by the promotion of the use of the TARGET Toolkit resources, specifically the use of the TARGET Antibiotic Checklist for patients who present with an antibiotic prescription and the use of the TARGET “treating your leaflets” for UTIs (and RTIs) when responding to symptoms for walk-in patients. 

A study limitation is that the data were self-reported by community pharmacy teams without independent validation, so both inconsistent and/or inaccurate reporting cannot be ruled out. However, all (104,142) data entries were included in all of the analysis. 

Similar to the previous related studies, it could also be argued that pharmacy teams may behave differently to usual practice when incentivised, with only interim changes in behaviours. We cannot determine whether community pharmacy teams continued to use the TARGET TYI leaflets as the MYS tool only allowed a maximum of 15 data entries to be collected per community pharmacy participating. Additionally, patients who were signposted to other healthcare settings were not followed up. Patient outcome data was not collected and this would be a recommendation for any future studies.

Additionally, it is challenging to estimate the longer-term impact and sustainability of any of the AMS initiatives in the absence of national standards in community pharmacies to tackle antimicrobial resistance.

## 5. Conclusions

The study was conducted using data collected from most community pharmacies in England and over 100,000 patients. It is one of the first studies of its scale examining how community pharmacies respond to infection symptoms and identifies the most common symptoms reported by patients with UTIs, such as dysuria, increased frequency of urination, and increased urgency to urinate. These findings align with typical UTI symptoms. In addition, the study provides valuable information about the acceptability and impact of the use of the TARGET TYI UTI leaflets in the community pharmacy setting and the impact on behaviour and education to the public presenting with symptoms. Further, it highlights the importance of trained pharmacy staff in managing UTIs and providing appropriate advice to patients. 

The use of the TARGET TYI UTI leaflets as part of the PQS 2022–23 has supported community pharmacies responding to UTI symptoms in a structured way. The leaflets help to identify symptoms that are less likely to be a UTI and reassure patients, providing them with safety netting advice as well as promoting self-care whilst escalating those patients who require signposting to other healthcare settings. 

Community pharmacy teams were able to support the majority of patients with UTI symptoms, particularly (non-pregnant) women with none or only one symptom of dysuria, new nocturia, cloudy urine, and/or vaginal discharge, who could be counselled with self-care advice and supplied OTC remedies. The teams have also demonstrated their diligence and awareness in the identification of symptoms that should be assessed urgently by signposting patients to their GP or another healthcare setting using the recommendations in the TARGET TYI UTI leaflet. There is a need for community pharmacy teams to consistently promote the most evidence-based interventions when antibiotics are not appropriate, i.e., pain relief and D-Mannose can be suitable for acute symptoms of self-limiting UTIs. Promoting symptomatic relief using appropriate OTC remedies and reducing the use of antimicrobials supports AMS principles to decrease the development of AMR across England. 

This initiative builds on previous AMS criteria, which have been included in the PQS since 2020. Together, these initiatives demonstrate how community pharmacy teams in England are supporting primary care and managing patients with common infections using a variety of resources from the TARGET Toolkit. TARGET resources were originally developed by the UKHSA (previously known as PHE), in collaboration with the Royal College of General Practitioners (RCGP), and the Antimicrobial Stewardship in Primary Care (ASPIC) Group to support GPs and other primary care healthcare professionals and commission organisations to improve antibiotic prescribing. Subsequently they have been modified to support their specific use in community pharmacies whilst ensuring that patient messaging across all resources for all healthcare professionals remains consistent.

Collectively, the use of TARGET resources and other AMS initiatives within the PQS have helped to educate both pharmacy teams and patients on AMS with the aim of tackling the development of AMR across England. Further studies to evaluate the impact of the health outcomes of patients included in a similar study would be useful. Community pharmacists and their teams play an increasingly important role in educating patients and the public about the importance of appropriate antimicrobial use, the risks of resistance and the significance of reducing the emergence and spread of AMR, which in turn promotes patient safety and improves the overall quality of patient care.

Embedding the use of TARGET resources into everyday practice within community pharmacies in England has potentially helped support them in managing patients’ personal attitudes, social norms, and to overcome perceived barriers to responsible antibiotic use. The collective initiatives have allowed community pharmacies to develop a solid foundation in their readiness for the launch of the Common Conditions Service where AMS considerations will be imperative in managing service users effectively.

## Figures and Tables

**Figure 1 antibiotics-12-01383-f001:**
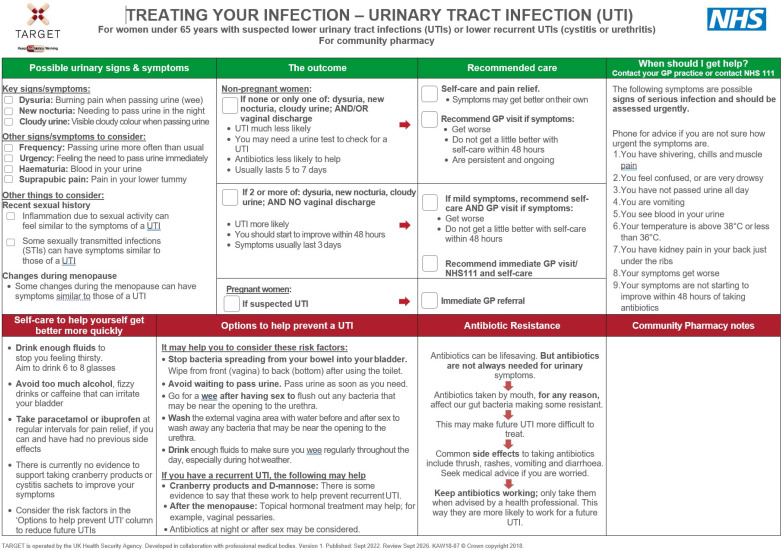
The TARGET Treating Your Infection Urinary Tract Infection patient information leaflet for community pharmacy settings.

**Table 1 antibiotics-12-01383-t001:** Patient symptom presentation to the pharmacy team.

TYI Leaflet “Possible Urinary Signs and Symptoms” *	Number of Patients (%)
Dysuria	55,665 (53)
(Increased) Frequency	53,609 (51)
(Increased) Urgency	49,397 (47)
Cloudy urine	29,899 (29)
Suprapubic pain	16,816 (16)
New Nocturia	14,839 (14)
Haematuria	6285 (6)
Abnormal vaginal discharge	5935 (6)
Other mainly comprising of:	2048 (2)
- back pain	473 (0)
- vaginal itching	385 (0)
- smelly urine	211 (0)
**Total**	**104,142**
**Number of presenting symptoms**	
1	33,428 (32)
2–3	55,805 (54)
4–5	13,422 (13)
>5	1487 (1)
**Total**	**104,142 (100)**
**Strongly predictive: Two or more of: cloudy urine, nocturia, dysuria (non-pregnant patient)**	22,381 (23)

* Multiple symptoms could be ticked simultaneously.

**Table 2 antibiotics-12-01383-t002:** Patient escalation symptoms and referrals to healthcare settings.

TYI Leaflet ‘When Should I Get Help?’ Escalation Symptoms **	Number of Escalation Symptoms from Total Patients (%)	Number of Escalation Symptoms from Patients with 2 or More UTI Symptoms * (%)
	**Number of Symptoms (%)**	
Symptoms getting worse	6788 (29)	2489 (32)
Blood in urine	4707 (20)	1397 (18)
Kidney pain in back, just below the ribs	3897 (17)	1376 (17)
Other:	2232 (9)	470 (6)
—Gestational Diabetes	1004 (4)	n/a
—Other	1228 (5)	n/a
Shivering chills and muscle pain	1782 (8)	706 (9)
Temperature >38 °C or <36 °C	1549 (7)	615 (8)
No improvement within 48 hrs of taking antibiotics	1130 (5)	424 (5)
Confusion or very drowsy	512 (2)	213 (3)
Not passed urine all day	368 (2)	109 (1)
Vomiting	198 (1)	75 (1)
**Total**	**23,163 (100)**	**7874 (100)**

* These are non-pregnant women with symptoms strongly predictive of a UTI (presenting with 2 or more symptoms of dysuria, new nocturia, cloudy urine, and with no vaginal discharge); ** Multiple symptoms could be ticked simultaneously.

**Table 3 antibiotics-12-01383-t003:** Signposting timescales to other healthcare services.

Type of Signposting (as per TARGET Leaflet Recommendations)	Number of Patients (%)	Patients with 2 or more UTI Symptoms * (%)
For urgent assessment	15,846 (15)	4928 (22)
If symptoms did not improve within 48 h	4597 (4)	1266 (6)
If symptoms got worse	2464 (2)	564 (3)
N/A (not referred to other services)	444 (0)	125 (1)
**Total number of patients**	**23,351**	**6883**
**Patient escalation to other health care settings based on leaflet recommendations**		
Yes	23,351 (22)	6883 (31)
No	80,791 (78)	15,498 (69)
**Total number of patients**	**104,142**	**22,381**

* These are non-pregnant women with symptoms strongly predictive of a UTI (presenting with 2 or more symptoms of dysuria, new nocturia, cloudy urine, and with no vaginal discharge).

**Table 4 antibiotics-12-01383-t004:** Over the counter (OTC) remedies recommended by patients.

Over the Counter Treatments **	Total Patients (%)	Patients with 2 or more UTI Symptoms (%) *	Patients with ‘When Should I Get Help?’ Escalation Symptoms (%)
Cystitis relief sachets	53,583 (76)	9792 (70)	1867 (46)
Pain relief	27,533 (39)	7541 (54)	2926 (72)
Cranberry products	10,034 (14)	2010 (14)	441 (11)
D-mannose	788 (1)	178 (1)	36 (1)
Other	4869 (7)	1456 (10)	167 (4)
—Clotrimazole products	1420 (2)	n/a	n/a
—Nitrofurantoin (via PGD)	748 (1)	232 (2)	n/a
**Total**	**70,466 (100)**	**13,994 (100)**	**4061 (100)**

* These are non-pregnant women with symptoms strongly predictive of a UTI (presenting with 2 or more symptoms of dysuria, new nocturia, cloudy urine, and with no vaginal discharge); ** Multiple medicines could be supplied simultaneously.

## Data Availability

The data presented in this study are available on reasonable request from the corresponding authors.

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
