# Peer review of "Responding to Urinary Tract Infection Symptoms in England’s Community Pharmacies"

_antibiotics, 2023, doi:10.3390/antibiotics12091383_

Round 1

Reviewer 1 Report

Dear authors,

Having reviewed your paper I have the following comments

Major comments

-As it stands it looks like this NHS treatment recommendation has been applied by pharmacies without questioning the evidence for it. If the evidence is available, it should be mentioned in the paper.

-it is recognised in the paper itself that some OTC medication is unlikely to have worked on an acute event. The study should have predefined symptomatics/pathogenics that are expected to work in the said condition.  Otherwise a placebo tablet (with no AEs) would have sufficed.

-line 54-55: when is a UTI seen by the pharmacist considered not resolved?

-line 156: why was a pregnancy test (available in the pharmacy) not employed?

-was fever ever considered and what happens with a patient that has fever, dysuria, and haematuria?

-how does the pharmacy staff differentiate between a lower and a higher UTI? It is noteworthy that the vast majority of patients had 2 or more symptoms making an UTI more likely 9although only around 1/4 had symptoms strongly predictive). How do we know how many of these were uncomplicated low UTIs and how is the diagnosis made in the pharmacy?

-importantly: was there any follow-up of patients? How does the pharmacy know how many patients handled directly had to use another service provider on their own motion?

-line 308: what is the exact evidence of efficacy and how was this evaluated in this study?

-in view of the above I cannot agree to the conclusions. 

Minor comments

-on line 201: the percent is 23 and not 22

-line 262: cannot understand this percent (15%). Please explain

Author Response

Dear Reviewer,

thank you for your feedback - please see our response attached.

Reviewer 2 Report

This study was aimed to promote antimicrobial stewardship (AMS) and decrease antimicrobial resistance. Despite the important clinical implications of this study, the authors must address some areas of concern.

Areas of concern:

Introduction

The authors could have briefly presented the prevalence of UTIs in England.

Materials and methods

From the background (introduction), we understand the focus of this study on women but the authors did not explain the reason underlining the restriction to women aged 65 years or under.

Results

Line 177: Write ‘‘not meet’’ instead of ‘‘not met’’

Discussion

The authors clearly indicated that the use of TARGET resources and other AMS initiatives within the PQS have helped to educate both pharmacy teams and patients on AMS to decrease the development of AMR across England. However, by extrapolation, the authors failed to point out that the use of these OTC medicines for the management of uncomplicated UTIs could also help the patients to save money would be used for antibiotics.

Conclusion

The authors failed to recommend further studies to evaluate the impact of such a study on the health outcomes of the participating patients.

Only one grammatical issue was detected (See line 177).

Author Response

Dear Reviewer,

Thank you for your feedback, please see our response attached.

Reviewer 3 Report

In this article authors used data of TARGET Treating Your Infection (TYI) leaflets, used by community pharmacy staff during consultations 14 whilst responding to UTI symptoms for women 65 years or younger.

1. It is not clear, why only female patient aged 65 years was eligible for inclusion?

2. Author has not mentioned the exclusion criteria also.

3. The article lacks novelty. 

---------------

Author Response

Dear Reviewer,

Thank you for your feedback, please find our response attached.

Reviewer 4 Report

Line 14-15: Please restructure the sentence 

Line 32:  94% (97,452) of women….. A total of 94%

Line 271: Authors are requested to elaborate and discuss the role of pharmacies as ‘antibitoic guardians’ like trainings imparted, applications of tests like antibiotic susceptibility testing by these pharmacies or any other lab etc.

The results presented in the study lack detailed information about patient demographics, authors may add (if available) this part. The additional data would provide a more comprehensive understanding about findings.

General comment:

The study was conducted using data collected from ample number of community pharmacies and patients making sample size fit for substantial for analysis. The study identifies the most common symptoms reported by patients with UTIs, such as dysuria, increased frequency of urination, and increased urgency to urinate. These findings align with typical UTI symptoms reported in the literature.  The study provides valuable information about the usage of the TARGET TYI UTI leaflets. This highlights the importance of properly trained pharmacy staff in managing UTIs and providing appropriate advice to patients.

Author Response

Dear reviewer,

thank you for your feedback, please find our response attached.

Reviewer 5 Report

This is a well-designed study and a well-written manuscript. No significant issue has been detected.

Author Response

Comments and Suggestions for Authors

This is a well-designed study and a well-written manuscript. No significant issue has been detected.

Thank you for your feedback.

Round 2

Reviewer 1 Report

Dear editor,

Having reviewed the changes proposed by the authors, which obviously cannot possibly address all shortcomings, especially the methodological ones, I can recommend its publication, albeit a bit reluctantly

Reviewer 3 Report

NA